# The Impact of Pulmonary Vein Anatomy on P-Wave Appearance during Sinus Rhythm: Cardiac Computed Tomography Study

**DOI:** 10.3390/diagnostics13182911

**Published:** 2023-09-11

**Authors:** Viktorija Verhovceva, Ligita Zvaigzne, Romans Lācis, Oskars Kalējs

**Affiliations:** 1Department of Internal Diseases, Pauls Stradins Clinical University Hospital, 13 Pilsoņu Str., LV-1002 Rīga, Latvia; ligita.zvaigzne@inbox.lv (L.Z.); romans.lacis@rsu.lv (R.L.); okalejs@gmail.com (O.K.); 2Department of Internal Diseases, Faculty of Medicine, Rīga Stradins University, 16 Dzirciema Str., LV-1007 Rīga, Latvia

**Keywords:** cardiac computed tomography, electrocardiography, pulmonary veins, P-wave, left atrial appendage

## Abstract

Electrocardigraphy remains a first-line evaluation method for cardiac electrical activity, recorded from the body surface. Since atrial activation is seen on the ECG as a P-wave, several factors are known to impact the appearance of the P-wave, such as the direction of electric impulse, conduction abnormalities, and anatomical characteristics of the atria. This retrospective study aimed to find statistically significant associations between the anatomy of pulmonary veins (PVs) observed in cardiac computed tomography (CT) and P-wave appearance during sinus rhythm on resting ECG. For each patient, a resting 12-lead ECG was recorded, and the field of analysis was P-wave—its duration, morphology, and axis. The evaluation of the CT scan recordings was performed by creating 3D models of the left atrium and analyzing the anatomy of the PVs and left atrial appendages (LAA). Noteworthy correlations were found: anatomy of the left PVs showed an association with LAA volume, LAA morphology, and P-wave notching in lead II. The right PVs demonstrated a relation with the P-wave axis and amplitude. Although these correlations cannot be classified as strong, the results not only expand understanding about discussed variables but also suggest the presence of a subtle and complex relationship, that warrants further exploration.

## 1. Introduction

The electrocardiograph was invented in 1902 by Dutch physiologist Willem Einthoven [1], starting a new epoch of cardiac electric impulse investigation for diagnostic purposes. This invention laid the groundwork for electrocardiography as it is known nowadays. There are many reasons for electrocardiography to remain a first-step evaluation method of cardiac pathology in the era of digital technology, such as its simplicity and quickness of performance, cost-effectiveness, and non-invasiveness. The purpose of an electrocardiogram (ECG) is to record electric activity in the heart from the surface of the body, representing the results as a graph with impulse voltage on the vertical axis and time on the horizontal axis. The graphical representation of electrical impulse provides an opportunity to examine its direction, intensity, and distribution in the tissue, recorded from different views in frontal and horizontal planes [2]. 

To create a dimensional view, a 12-lead ECG is used: three bipolar limb leads (I, II, III), three augmented limb leads (aVR, aVL, aVF), and six chest leads (V1–V6). According to international nomenclature, the P-wave represents depolarization of the atria, the QRS complex shows electric activity in the ventricles and the T wave describes ventricular repolarization [3]. The normal atrial depolarization impulse is generated in the atrial pacemaker complex in or near the sinoatrial node, further activation spreads anteriorly and inferiorly to the bottom part of the right atrium (RA) with subsequent propagation to the left atrium (LA). Usually, the LA activity continues after the end of RA depolarization. These activation patterns in the ECG are seen as the P-wave. A normal P-wave is positive in lead I, II, aVL, and aVF, representing the direction of atrial activation, with a mean P-wave axis in the frontal plane between 0° and +75° [2,4]. From the horizontal plane perspective, the first electric impulse shift goes anteriorly, then leftward and posteriorly, reflected as an upright P-wave in precordial leads, with possible biphasic morphology in leads V1 and V2 [4]. Based on the depolarization spreading pattern in the atria, the first part of the P-wave represents activity in RA, the middle part represents the ending of RA depolarization and initiation of LA activity, and the terminal part represents the ending of LA activation [2]. A normal P-wave duration is suggested to be 120 msec, with limb lead amplitude less than 0.25 mV and the depth of the final parts’ negative component in leads V1–V2 is less than 0.1 mV [4].

The body surface ECG reveals only major trends of atrial excitation since only the currents transmitted to the electrode placement area can be recorded. Additionally, the ECG represents a summation of the electrical potentials generated by countless numbers of cells [5,6]. However, knowing the basics of the normal ECG, the morphology of the P-wave provides an opportunity to assess various atrial pathologies, including right atrial abnormalities, left atrial abnormalities, and interatrial conduction disorders [6].

Typical ECG findings in right atrial (RA) pathology consist of tall, peaked P-waves in limb and precordial leads, with an amplitude of 0.25 mV or higher (Figure 1B). This is because the depolarization forces in the RA are primarily seen in the early part of the P-wave, and isolated RA hypertrophy does not typically cause significant P-wave prolongation. Electrophysiology studies have shown that patients with tricuspid valve disease, chronic cor pulmonale, or congenital heart disease commonly exhibit a P-wave axis in the frontal plane of 75 degrees or higher, along with a positive deflection of the P-wave in lead V1 or V2 exceeding 0.15 mV. However, other findings associated with RA abnormalities have a poor correlation with anatomical findings [7]. For example, out of one hundred patients clinically diagnosed with cor pulmonale, only twenty will display the typical ECG pattern with peaked P-waves [8,9,10]. According to a study by Shleser et al. [11], tall, peaked P-waves can also be present in healthy individuals with an asthenic build and may be attributed to the vertical position of the heart. In conditions characterized by predominant sympathetic stimulation and increased cardiac output, such as expiratory maneuvers against pressure, physical exercise, tachycardia, or in individuals with left pericardial defects, the amplitude of the P-wave can also be increased [12,13,14]. There is even a study describing increased voltage and right axis deviation of the P-wave in subjects with coronary artery disease [15].

Left atrial (LA) abnormality, often referred to as dilation or actual enlargement, leads to prolongation of the middle and terminal parts of the P-wave, with predominant leftward and posteriorly oriented electrical forces. Therefore, specific ECG findings associated with LA pathology include a notched P-wave with a duration exceeding 120 msec, a biphasic P-wave in lead V1 with a prominent terminal negative deflection longer than 40 msec and depth greater than 1 mm, as well as a leftward shift of the P-wave axis in the frontal plane of 15 degrees or more (Figure 1C) [5,6]. In a study conducted by Saunders et al. [16] on 62 patients with significant mitral stenosis and LA enlargement, specific ECG findings for LA pathology were observed in only 62% of cases, which aligns with similar results reported in other studies [17]. An anatomical correlation study by Romhilt et al. [18] revealed that the terminal deflection of the P-wave correlates more with the volume of the LA rather than the LA pressure. P-wave prolongation is observed in approximately two-thirds of subjects with clinically proven LA enlargement [17,18]. Insignificant notching of the P-wave can be seen in ECGs of healthy individuals, more frequently in precordial leads, typically with a peak-to-peak distance of less than 40 msec [19]. P-wave axis deviation is observed in only 10% of patients with surgically proven LA enlargement and in 5% of healthy individuals [5]. In an echocardiography study by Chirife et al. [20], P-wave duration criteria demonstrated a specificity of 89%. However, in another study, a combination of several ECG criteria showed a predictive index for the clinical presence of LA enlargement in 63% of cases and 78% for the absence of LA pathology [21]. The biphasic P-wave is described as specific (92%) but with a low sensitivity score of 12% [22].

**Figure 1 diagnostics-13-02911-f001:**
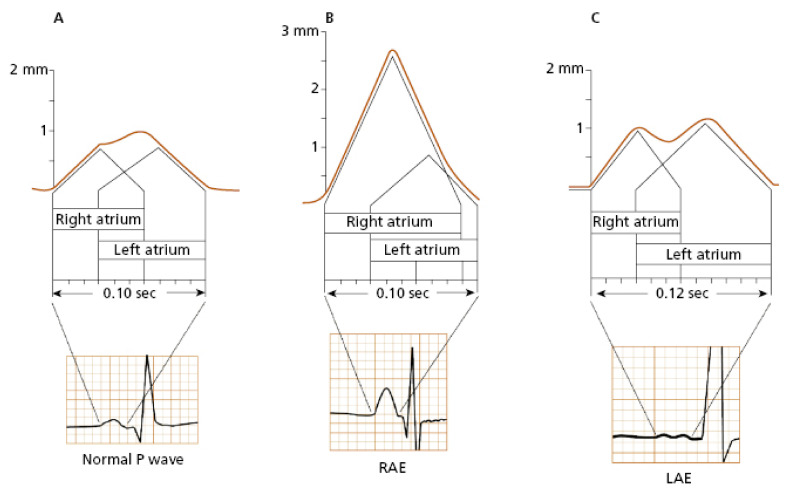
The visual representation of the atrial depolarization in a normal P-wave (**A**), right atrial enlargement (**B**), and left atrial enlargement (**C**) [23].

In cases where the P-wave is prolonged, with or without notching, and exhibits low or normal amplitude, while lacking indirect or clinical signs of left atrial (LA) enlargement, the most appropriate ECG conclusion would be interatrial or intraatrial conduction abnormalities [5]. Notching of the P-wave can also be observed in individuals with atrial ischemia, infarction, fibrosis, or myocarditis. In a prospective study, a prolonged P-wave duration exceeding 120 msec was observed in 40.6% of hospitalized patients [24]. Since our study is focused on the correlation between ECG findings and the anatomical observations from cardiac CT, no further conduction analysis was made.

As observed in the previous analysis, although the electrocardiogram (ECG) may not offer precise information about certain cardiac conditions, it continues to possess significant clinical value for several reasons. Firstly, electrocardiography is a non-invasive and relatively inexpensive procedure when compared to other diagnostic methods like cardiac catheterization or imaging techniques. Additionally, it is a readily accessible tool that can be swiftly performed at the bedside or in an outpatient setting. Moreover, the ECG functions as an initial screening tool for detecting various heart abnormalities and rhythm disturbances. The information derived from an ECG can effectively guide healthcare professionals in making decisions regarding further diagnostic testing and determining appropriate treatment strategies.

This study aimed to find statistically significant correlations between the characteristics of the P-wave on the ECG during sinus rhythm and the anatomy of the pulmonary veins (PVs) found in computed tomography (CT) scans. Despite the fact, that the area around PVs is known to be an electrically active structure, there are limited data on the analysis of associations between the anatomy of PVs and ECG findings. The muscular sleeves extended around the PVs are known to provide pacemaker activity, which was first described in a guinea pig model [25] and later immunohistochemistry study by Blom et al. [26] proved transitory HNK-1 antigen expression around the PVs, which is associated with the development of the cardiac impulse conduction system in the model of the human embryo [27]. Numerous human mapping studies have shown pacemaker foci in the proximal part of the PVs [27,28,29], ablation of which may result in the disappearance of arrhythmias [27]. 

The clinical significance of our study is underscored by the findings from Haïssaguerre et al. [28], revealing that 94% of the ectopic impulses initiating atrial fibrillation (AF) originate from the periphery of the pulmonary veins (PVs). With the global prevalence of AF having notably surged in recent decades, currently standing at approximately 60 million cases worldwide [30], there exists a compelling trend toward further escalation. Therefore, atypical atrial depolarization leading to arrhythmogenesis is a rapidly expanding field of research [4,24,31]. AF is a result of a complex pathogenetic process with serious consequences, which can potentially be mitigated through timely intervention. It is crucial to emphasize that nearly 30% of all thromboembolic events can be attributed to AF, and around one-third of AF patients remain asymptomatic [32]. The P-wave morphology has been investigated from numerous perspectives in clinical practice, describing its role in the prediction of clinical AF [33], recurrence of AF after electrical cardioversion [34], and even the detection of left ventricular diastolic dysfunction [35].

In clinical practice, the most precise method for evaluation of the electrical activity of the heart is electroanatomical mapping (CARTO, RHYTHMIA, EnSite systems), which is an invasive technique that integrates real-time electrical measurements with high-resolution three-dimensional anatomical visualization [36]. The invasivity of the procedure is often associated with increased risk for infections, longer recovery time, and require special personnel training. 

This study describes the impact of anatomical variations of the PVs on the morphology of the ECG P-wave, showing the electrical activity of the PVs in a non-invasive way, noticeably reducing the risks associated with invasive manipulations. The obtained data may be useful in several aritmology perspectives, such as improved understanding of arrhythmogenesis with potential benefit value in risk stratification, novel biomarker identification, optimization of the diagnostic algorithms, and enhancement of the therapeutical approach by targeted therapy planning, individualized treatment, and long-term follow-up strategies. 

## 2. Materials and Methods

In this retrospective study, 145 patients aged 18 years and older, without previously registered cardiovascular pathology, were enrolled. All patients were evaluated by following anthropometric criteria—age, sex, weight, and height. Right after the measurement of anthropometric data, a resting 12-lead ECG was recorded, strictly 15 min into resting position. The ECG was recorded at a speed of 25 mm/s with a voltage of 10 mm/mV. Subsequently, a multi-slice cardiac computed tomography (CT) scan was performed with the patient in a supine position, preceded by the administration of beta-blocker medication to maintain the heart rate in the range of 60–70 beats per minute.

All the ECG and CT scan recordings were analyzed manually. In the ECG, the following criteria were evaluated—rhythm, rate, P-wave duration, amplitude, axis, presence of notching in lead II, and morphology in lead V1. In the CT scan recordings, the LA was the field of analysis, performed using five three-dimensional (3D) reconstruction models of the heart during the cardiac cycle for each patient. The phases of the cardiac cycle were evaluated based on the ECG registered during the cardiac CT. The 3D LA models were used to analyze the number of the PV orifices on the posterior wall of the LA, the distance between PV orifices on each side (in millimeters), the angle between PVs (left and right, in degrees), and statistics of the left atrial appendage (LAA)—volume (in milliliters) and morphology (chicken wing, cactus, cauliflower, windsock).

Statistical evaluation of the data was made using the IBM SPSS Statistics 27 program, with a statistical significance value of *p* < 0.05. Approval from the Medical Ethics Committee was obtained prior to the study, and all scientific investigations were conducted in accordance with the Declaration of Helsinki.

## 3. Results

A population of 145 patients with no previously known cardiovascular pathology was enrolled in the study, with a mean age of 67.0 years (SD = 7.9). Out of the total, 75.2% were male (109) and 24.8% (36) female. 

The mean P-wave duration was 0.113 s (SD = 0.017), the mean amplitude was 0.146 mV (SD = 0.061), and the mean P-wave axis was 58.850° (SD = 31.784) (Table 1). The presence of P-wave notching was seen in 35.7% (43) of cases. In lead V1, the P-wave showed different morphologies: biphasic with similar depth of both phases in 35.2% (43) of cases, biphasic with a larger negative phase in 21.3% (26), biphasic with a flat positive first phase in 16.4% (20), biphasic with a smaller negative phase in 9.8% (12), monophasic positive in 10.7% (13), monophasic negative in 3.3% (4), and biphasic with a flat negative second phase in 3.3% (4) of cases (Table 2). 

The most common variation observed was two right PVs (2 orifices) seen in 86.2% of cases (125), followed by three PV orifices 11.7% (17), four PV orifices 1.4% (2), and one PV orifice 0.7% (1) (Figure 2). On the left side of the LA posterior wall, two PV orifices were seen in 77.2% (112) of cases, and one PV orifice in 22.8% (33) (Figure 3, Table 3). The mean distance between right PVs was 8.2 mm (SD = 3.8) and between left PVs 5.9 mm (SD = 3.2). The mean angle between the right PVs was 50.16° (SD = 14.68) and between the left PVs was 48.5° (SD = 13.5) (Table 4). 

The most common morphology of the LAA was chicken wing in 65.5% (95) of cases, followed by cactus morphology in 17.9% (26), cauliflower in 13.1% (19), and windsock in 3.4% (5) of cases (Figure 4, Table 5). The mean volume of the LAA was 12.1 mL (SD = 4.8) (Table 6).

A statistically significant negative correlation was found between the left PV angle and the volume of the LAA (rs = −0.269, *p* = 0.002). However, the right PV angle showed no significant correlation with the LAA volume (*p* = 0.678) (Table 7). The left PV angle exhibited statistically significant differences among LAA morphology groups (*p* = 0.019) (Figure 5), while no significant difference was found in the right PV angle among LAA morphology groups (*p* = 0.688). The number of left PV orifices showed a statistically significant distribution difference among P-wave notching groups found in lead II (*p* = 0.026). In cases with one PV orifice, notched P-waves were seen in 3.45% (5) of cases without notching in 15.17% (22), whereas in cases with two PV orifices, notched P-waves were seen in 25.52% (37) and without notching in 39.31% (57) (Table 8). There was a positive correlation between the right and left pulmonary vein angle (rs = 0.325, *p* < 0.001).

The number of right PV orifices showed a positive correlation with the P-wave axis (rs = 0.239, *p* = 0.008). Additionally, the distance between right PV orifices had a negative correlation with P-wave amplitude (rs = −0.193, *p* = 0.047).

## 4. Discussion

Our study demonstrated statistically significant associations between parameters of the pulmonary veins and the distribution of electrical impulses in atrial tissue, as observed on the ECG. Among the significant findings, a negative correlation was found between the left PV angle and the volume of the LAA (rs = −0.269, *p* = 0.002), and statistically significant differences were observed in the LAA morphology groups (*p* = 0.019). These findings could potentially be explained by embryonic development effects, considering that the right PV angle did not have any impact on LAA volume (*p* = 0.678) and showed no distribution differences among LAA morphology groups (*p* = 0.688). Found relationships suggest that local changes during the growth and formation of LA structures, particularly in relation to the proximity of the left PVs to the LAA, may play a role. Further studies in this direction could provide valuable insights.

Another notable finding is the distribution difference in the number of left PV orifices among P-wave notching groups found in lead II (*p* = 0.026). Specifically, among all cases, notched P-waves were associated with the presence of two left PV orifices in 37 (25.52%) cases, whereas in hearts with a single common left PV, notching of the P-wave was observed in only five cases (3.45%). Given the statistical significance with a *p*-value of 0.026, this difference cannot be solely explained by the fact that two left PV orifices were more frequently observed than a single common PV. A similar numerical proportion of PV ostia, as described in our study, was also observed in the research conducted by Wittkampf et al. [38], where in a group of 42 patients undergoing an MR study, only four patients had a single common left PV connecting to the posterolateral wall of the LA.

There was also a positive correlation between the right and left PV angles (rs = 0.325, *p* < 0.001). Although this anatomical difference may initially seem clinically insignificant, it can have practical implications in the field of interventional medicine. These findings could be particularly useful for pre-interventional planning or predicting postprocedural outcomes. From a genetic and molecular profiling perspective, the identified associations could be further analyzed to identify potential biomarkers or genetic markers that could aid in risk stratification and targeted treatment strategies.

The number of right PV orifices showed a statistically significant positive correlation with the P-wave axis (rs = 0.239, *p* = 0.008), while the distance between right PV orifices exhibited a negative correlation with P-wave amplitude (rs = −0.193, *p* = 0.047). These associations may be related to the stretching motion of the atrial wall during the cardiac cycle, as blood flow through the PVs creates unevenly distributed wall stress. Consequently, the aforementioned findings could be influenced by blood pressure, which presents a promising avenue for further studies.

For future investigations in this direction, several improvements could be considered. Firstly, using longer ECG recordings such as ambulatory Holter monitoring could reveal transient impulse conduction abnormalities. Lowering the speed of ECG recording and adjusting voltage calibration settings may unveil more subtle conduction features. Additionally, expanding the study to include patients with arrhythmia and switching from resting ECG to stress ECG could uncover unexpected conduction abnormalities. Three-dimensional electroanatomical mapping systems could be utilized in further studies to create detailed maps of the electrical activity of the PVs, thus identifying precise locations of electrical impulses originating from the PV area.

It should be acknowledged that the associations discovered in this study, as indicated by Spearman’s rank correlation coefficient, cannot be classified as strong; however, the findings remain significant in numerous aspects. Firstly, the identified relationships enhance our understanding of the impact of PV anatomy on ECG findings, suggesting the presence of subtle and complex connections that warrant further investigation. The data obtained from this study contributes to balancing the existing knowledge in the field. Conducting studies, even if they yield inconclusive results, is crucial in preventing the propagation of biases and false conclusions. Research is a collective endeavor, with each study contributing to a broader body of knowledge. Even if an individual study does not yield strong correlations, its findings can be integrated into the existing literature and contribute to a more comprehensive understanding of a particular field. This cumulative knowledge serves as the foundation for future research and advancements.

In summary, this study has successfully established significant correlations between electric activity and anatomical parameters of the human heart. While the associations may not be classified as strong, they provide valuable insights and contribute to the ongoing progress in the field. Further research building upon these findings will continue to advance our understanding of the complex relationships between the electrical and anatomical aspects of the heart.

## 5. Conclusions

In conclusion, the morphology of the P-wave during sinus rhythm is the result of a complex interaction between the anatomical, geometrical, and electrophysiological characteristics of the atria, with the specific role in P-wave appearance often being uncertain. However, this study has demonstrated statistically significant associations between the left pulmonary veins and characteristics of the left atrial appendage, such as morphology and volume. The number of left pulmonary veins has also been shown to be related to the morphology of the P-wave in lead II, indicating an impact on P-wave notching. Furthermore, the pulmonary veins on the right side have shown statistically significant correlations with P-wave axis and amplitude, highlighting a broad field for further investigations. These findings contribute to our understanding of atrial anatomy and function and have implications for clinical practice, pre-interventional planning, and future research in the field of cardiac electrophysiology.

## Figures and Tables

**Figure 2 diagnostics-13-02911-f002:**
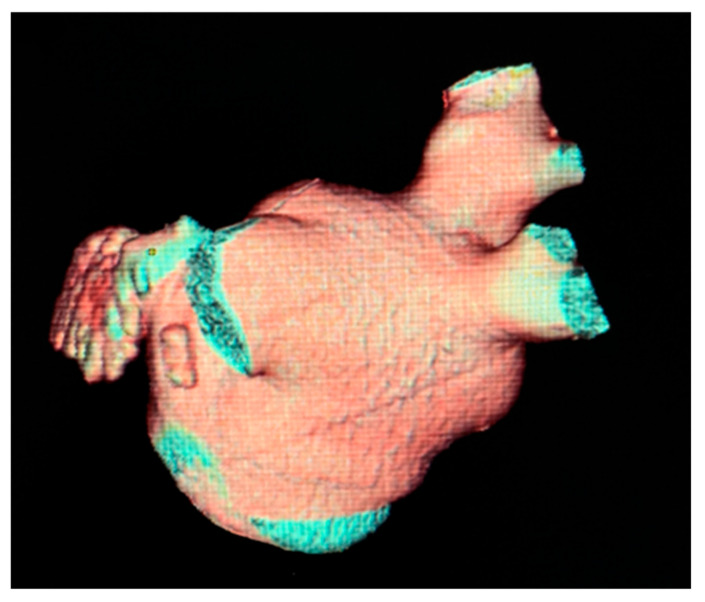
Cardiac CT 3D reconstruction of the left atrium. Posterolateral view—one common pulmonary vein on the left side [37].

**Figure 3 diagnostics-13-02911-f003:**
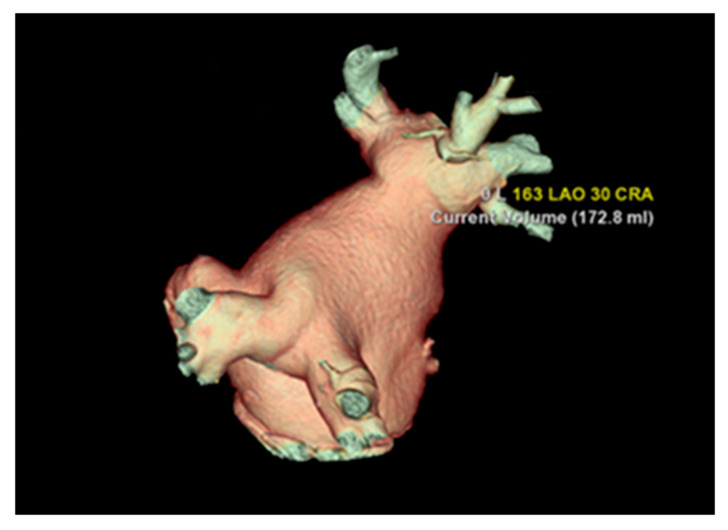
Cardiac CT 3D reconstruction model of the left atrium. Posterolateral view—one common pulmonary vein on each side.

**Figure 4 diagnostics-13-02911-f004:**
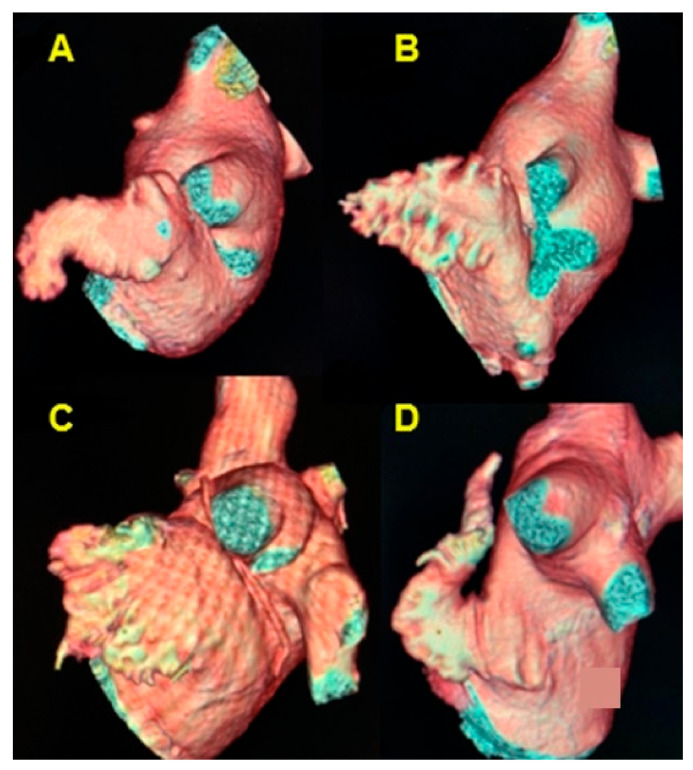
Cardiac CT 3D reconstruction model of the left atrium. Lateral view, the morphology of the left atrial appendage. (**A**) chicken wing; (**B**) cactus; (**C**) cauliflower; (**D**) windsock.

**Figure 5 diagnostics-13-02911-f005:**
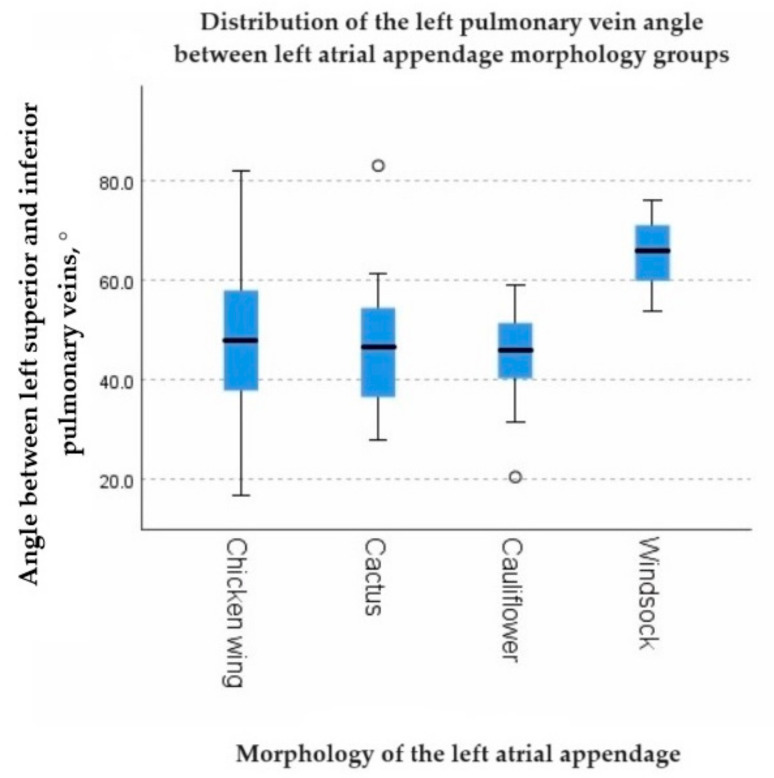
The graph showing the distribution of the left pulmonary vein angle between left atrial appendage morphology groups.

**Table 1 diagnostics-13-02911-t001:** P-wave statistics.

P-Wave	Mean Value	SD
duration, s	0.113	SD = 0.017
amplitude, mV	0.146	SD = 0.061
axis, °	58.850	SD = 31.784

SD, standard deviation.

**Table 2 diagnostics-13-02911-t002:** P-wave morphology.

P-Wave	Frequency, *n*	%
notching in lead II	43	35.7
morphology in lead V1		
biphasic with a similar depth of both phases	43	35.7
biphasic with a bigger negative phase	26	21.3
biphasic with a flat positive first phase	20	16.4
biphasic with a smaller negative phase	12	9.8
monophasic positive	13	10.7
monophasic negative	4	3.3
biphasic with a flat negative second phase	4	3.3

P-wave notching, M-shaped P-wave morphology with peak-to-peak distance ≥ 0.04 s.

**Table 3 diagnostics-13-02911-t003:** Number of pulmonary vein orifices.

Number of PV Orifices	Prevalence, *n*	%
Right PVs		
two orifices	120	85.7
three orifices	17	12.2
four orifices	2	1.4
one orifice	1	0.7
Left PVs		
two orifices	110	78.6
one orifice	30	21.4

PV, pulmonary vein.

**Table 4 diagnostics-13-02911-t004:** Statistics of pulmonary vein orifices.

Pulmonary Vein	Mean Value	SD
Distance between PV orifices, mm		
Right PVs	8.2	3.8
Left PVs	5.9	3.2
PV angle, °		
Right	50.16	14.68
Left	48.5	13.5

PV, pulmonary vein.

**Table 5 diagnostics-13-02911-t005:** Morphology of the left atrial appendage.

LAA Morphology Group	Prevalence, *n*	%
Chicken wing	95	65.5
Cactus	26	17.9
Cauliflower	19	13.1
Windsock	5	3.4

LAA, left atrial appendage.

**Table 6 diagnostics-13-02911-t006:** Volumetric parameters of the left atrial appendage.

LAA Volume	Value, mL	SD
Mean volume	12.1	4.8

LAA, left atrial appendage.

**Table 7 diagnostics-13-02911-t007:** Pulmonary vein angle correlation with the volume of the left atrial appendage (Spearman’s rank-order correlation test).

Parameters	r_s_	*p*
Right PV angle, °	0.035	0.678
Left PV angle, °	−0.269	0.002

PV, pulmonary vein.

**Table 8 diagnostics-13-02911-t008:** Distribution difference among P-wave notching groups in lead II.

Parameters	Frequency, N	%
one PV orifice		
notched	5	3.45
without notching	22	15.17
two PV orifices		
notched	37	25.52
without notching	57	39.31

PV, pulmonary vein.

## Data Availability

Not applicable.

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
