# Peer review of "The Impact of Pulmonary Vein Anatomy on P-Wave Appearance during Sinus Rhythm: Cardiac Computed Tomography Study"

_diagnostics, 2023, doi:10.3390/diagnostics13182911_

Round 1

Reviewer 1 Report

It is important to note the originality of the study. However, it is not clear why it was necessary to look for the discovered relationships. In the extensive topicality, the authors have not been able to problematize the studied issue. In addition, the bibliography does not appear to be up-to-date. It would be important to prove the existence of any associations between the ECG pattern and the results of cardiac CT under conditions of multiple comparison. Could the results of the study have been influenced by the introduction of beta-blockers, it is known that they affect the electrophysiology of the heart.

Author Response

We extend our sincere gratitude for your review of our manuscript and the kind advices that were taken into consideration during the editing process. The aim of our study was to identify statistically significant correlations between the characteristics of the P-wave on the ECG during sinus rhythm and the anatomy of the pulmonary veins (PVs) observed in computed tomography (CT) scans. While the most precise method for evaluating the heart's electrical activity is through invasive electroanatomical mapping, our study sought to establish a non-invasive approach for assessing the anatomical parameters of the PVs. The potential clinical significance of the data we obtained holds promise across various arrhythmology perspectives. This includes enhancing understanding of arrhythmogenesis for improved risk stratification, identifying novel biomarkers, optimizing diagnostic algorithms, and refining therapeutic approaches through targeted therapy planning, individualized treatment, and long-term follow-up strategies. The non-invasive nature of the examination significantly reduces the risks associated with invasive procedures. These risks encompass potential infections, patient discomfort, extended hospital stays, and the need for specialized personnel training. The manuscript has been reviewed and refined based on the latest relevant references. The introduction section includes a concise literature review, comparing studies conducted over the past few years. Regarding the possibility of ECG analysis being influenced by beta-blocker administration, it's important to note that the ECG was recorded prior to medication administration. The medication was administered just before the CT scan, as outlined in the Materials and Methods section.

Reviewer 2 Report

Nicely presented study. Although it's correlation isn't strong, it provides the ground for more studies around the subject in the future.

Author Response

We extend our sincere gratitude for your review of our manuscript. Your commitment to the peer-review process is profoundly valued, and we wish to express our deep appreciation for the feedback you have provided.

Reviewer 3 Report

The article makes the efoort of using ECG mophology of P wave in order to provide correlation with LA and PV morphology. There are a number of improvements that can be made: - in the abstract, one should replace the term "cardiac function" when describing the role of ECG, since ECG only reflects the conduction of the electrical signal in the heart, not the cardiac function; - the introduction is too long and it should not include references to results and conclusions; - figures could include P wave morphologies in different leads, instead of the general ECG aspect of figure 1; - tables 3 and 5 should replace the term "frequency" by "prevalence"; - discussion and abstract should more clearly present to what extend the findings in this article could help the clinician or the arrhythmologist for a therapeutical approach; - the references 36 and 37 should be properly written, with the full extent of their citation.

Author Response

We extend our sincere gratitude for your review of our manuscript.

  1. The term 'cardiac function' has been replaced with 'electrical activity.'
  2. The section containing results and conclusions in the introduction has been removed, as recommended. The introduction now outlines the study's purpose, the clinical importance of the obtained data, and a literature review related to the study's topic.
  3. Figure 1 has been replaced with a schematic representation of different P-wave morphologies, allowing readers to compare two distinct pathologies: right and left atrial enlargement.
  4. The term 'frequency' has been substituted with 'prevalence' in Tables 3 and 5.
  5. The revised introduction describes the clinical significance aspects of our study, including potential benefits for diagnostic and therapeutic strategies.
  6. References 36 and 37 have been corrected. Since Figures 3 and 4 are original images not previously published anywhere, the author and year are now provided.

Round 2

Reviewer 3 Report

Authors answered to reviewer's requests. There is only one place where 'frequency' was replaced by the same word, instead of using the term 'prevalence'